# Service Life Evaluation for RC Sewer Structure Repaired with Bacteria Mixed Coating: Through Probabilistic and Deterministic Method

**DOI:** 10.3390/ma14185424

**Published:** 2021-09-19

**Authors:** Hyun-Sub Yoon, Keun-Hyeok Yang, Kwang-Myong Lee, Seung-Jun Kwon

**Affiliations:** 1Department of Architectural Engineering, Kyonggi University, Suwon 16227, Korea; lonsohs@naver.com (H.-S.Y.); yangkh@kgu.ac.kr (K.-H.Y.); 2Department of Civil, Architectural and Environmental System Engineering, Sungkyunkwan University, Suwon 16419, Korea; leekm79@skku.edu; 3Department of Civil and Environmental Engineering, Hannam University, Daejeon 34430, Korea

**Keywords:** *Rhodobacter capsulatus*, sewage concrete, surface coating, sulfate ion, service life, cover depth

## Abstract

Since a concrete structure exposed to a sulfate environment is subject to surface ion ingress that yields cracking due to concrete swelling, its service life evaluation with an engineering modeling is very important. In this paper, cementitious repair materials containing bacteria, *Rhodobacter capsulatus,* and porous spores for immobilization were developed, and the service life of RC (Reinforced Concrete) structures with a developed bacteria-coating was evaluated through deterministic and probabilistic methods. Design parameters such protective coating thickness, diffusion coefficient, surface roughness, and exterior sulfate ion concentration were considered, and the service life was evaluated with the changing mean and coefficient of variation (COV) of each factor. From service life evaluation, more conservative results were evaluated with the probabilistic method than the deterministic method, and as a result of the analysis, coating thickness and surface roughness were derived as key design parameters for ensuring service life. In an environment exposed to an exterior sulfate concentration of 200 ppm, using the deterministic method, the service life was 17.3 years without repair, 19.7 years with normal repair mortar, and 29.6 years with the application of bacteria-coating. Additionally, when the probabilistic method is applied in the same environment, the service life was changed to 9.2–16.0 years, 10.5–18.2 years, and 15.4–27.4 years, respectively, depending on the variation of design parameters. The developed bacteria-coating technique showed a 1.47–1.50 times higher service life than the application of normal repair mortar, and the effect was much improved when it had a low COV of around 0.1.

## 1. Introduction

RC (Reinforced Concrete) sewage structures are a life-line system for the public life of users and require periodic repair and maintenance due to its deterioration. In general, RC structures are commonly used for sewage drainage systems, but the ingress of sulfate ions results in concrete surface layer deterioration, which yields reinforcement and leakage in the joints, thus incurring a large amount of cost to address these problems [1,2]. In the case of sulfate ingress, unlike the case of chloride attack and carbonation, cracking and disintegration from concrete expansion on the surface are major problems.

As for the bio-chemical deterioration of concrete, with the identification of Thiobacillus thiooxidans (the sulfate reducing bacteria), concrete surface erosion and the deterioration of internal hydrates have been reported [3]. Unlike the water supply pipes, the municipal wastewater flowing into the sewer has varying volume and velocity of inflow, and discharge in the pipe, and therefore, slime and sludge are usually deposited at the bottom of the duct. Anaerobic bacteria present in the depositions generate a large amount of hydrogen sulfide gas by reducing sulfate ions (SO_4_^2−^) to hydrogen sulfide (H_2_S) in the process of decomposing and consuming the deposited organic matter required as nutrients for growth [4]. This is due to the process of anaerobic respiration, in which anaerobic bacteria oxidize organic matter by using oxygen bonded with sulfur (SO_4_^2−^) instead of pure oxygen (O_2_) for the purpose of protein synthesis and the energy acquisition necessary for growth. In this process, hydrogen sulfide is generated, which has long been considered as the main deterioration factor of sewage facilities [5,6]. Several studies have started for the evaluation of concrete degradation due to sulfuric acid and hydrogen sulfide in sewage structures. These techniques can be used for indirect and quantitative methods to determine the propagation and growth level of sulfate-reducing bacteria. However, when using the procedure for determining the breeding and growth level of the bacteria, it is difficult to quantitatively evaluate the sulfate reducing bacteria due to various bacteria existing in the sewage environment; therefore, different technologies have been proposed for evaluating the concrete erosion progress by detecting changes in humidity and temperature in RC sewage structures considering the growth and reproduction of sulfate reducing bacteria [7,8]. In addition, techniques for predicting the depth of corrosion through monitoring the changing pH of surface concrete due to sulfuric acid and hydrogen sulfide gases formed by the growth of these bacteria have been proposed [9].

Figure 1 shows the metabolic process of generation of hydrogen sulfide due to the growth of anaerobic bacteria, and Figure 2 shows the deterioration mechanism of concrete caused by the effects of hydrogen sulfide and sulfuric acid.

In recent years, there has been active research on the development of repair techniques for protecting sewage structures and controlling deterioration factors, and most of the developed techniques are focused on crack healing and durability improvement of cementitious composites using bacteria [10,11,12]. These studies have shown the mechanisms in which calcium ions (Ca^+^) in cement mixtures decompose urea into ammonium and calcium carbonate (CaCO_3_) is formed through bacterial urea hydrolysis and the biomineralization of organic acid oxidation, thereby healing internal voids and microcracks. On the other hand, research on the use of bacteria with the function of inhibiting the penetration and diffusion of deterioration factors using a viscous slime formed on the cell surface have drawn attention. Studies have been conducted to form a surface protective coating on the concrete structures exposed to deterioration environments by culturing *Rhodobacter capsulatus*, a slime-forming bacterium [12,13,14]. A previous study showed that the coating repairing with bacteria demonstrated a superior sulfate resistance performance and showed an improvement in durability and mechanical performance such as reducing the diffusion of sulfate ions and maintaining strength at the exposure condition to sulfuric acid, compared to conventional concrete repair mortars [15]. When these repair materials are used in the RC sewage structure, the service life can increase, which reduces the maintenance cost, thus enabling the establishment of an effective maintenance plan. For a maintenance plan, an assessment of the RC structure through NDT (Non-Destructive Technique) such as 3D scanning and visual inspection is effective. As for cracks or surface disintegration in concrete due to chemical deterioration, an evaluation of the crack width and depth is necessary; however, the NDTs still have the limitation of application [16,17]. For the evaluation of cracks and large voids inside concrete, GPR (Ground Penetration Radar) is reported to be effective as well [18].

For the maintenance of the RC sewage structure, it is necessary to determine a limit state for service life, and a deterministic method is generally used. This method aims to secure the required performance within the intended service life considering the environmental factor and the durability reduction factor. In the specifications and guidelines of many countries, five deteriorations such as chloride attack, carbonation, sulfate ingress, freeze–thaw, and alkali aggregate reaction are considered as durability design items for RC structures [19,20,21].

For chloride attack and carbonation, a number of models for service life prediction have been proposed, which are based on a diffusion and convection mechanism. In the case of service life under sulfate attack, the deterioration depth equation [1,22] that takes into account the diffusion coefficient of sulfate ions and the mixing characteristics, service life evaluation considering the decrease in strength through acceleration test [23,24], and service life evaluation for multilayer diffusion considering coating diffusion have been studied [25]. The variation of chemical compositions and pH under sulfate ingress has been studied with lab-scaled, test, and corrosion depth, and weight loss has been related with duration period [9,26].

Many studies on the probabilistic service life evaluation have also been carried out in order to consider engineering uncertainties for quantitative materials, design, and construction. The probabilistic technique means a design method in which the probability of exceeding the critical condition during the intended service life is considered to be lower than the preset target durability probability [27,28,29,30]. For this purpose, the probability distribution of each design parameter should be determined, and the target durability probability (failure probability) according to the target service life should be defined by the design engineer. Since the 1990s, for chloride attack and carbonation, a number of design techniques have been proposed with deterministic and stochastic methods, and studies on analysis considering spatial variability reflecting time-dependent characteristics have been conducted [31,32]. However, there have been limitations in studies on the service life change in the case of repaired structures exposed to sulfate ingress such as RC sewage structures. This is because there may be a complicated deterioration process such as freeze–thaw, and cracks due to the expansion of the concrete and the consequent penetration of deterioration factors, and also sulfate ion ingress and reactions with internal hydrates (calcium hydroxide and gypsum) occur continuously. There have been several studies on sulfate-service life with deterministic methods, but there are very few studies on sulfate-service life considering probability theory and design parameters with variation.

Actually, durability design has developed from deterministic and probabilistic engineering uncertainties such as construction skill variation, design errors, and material quality variation. Regarding sulfate ion ingress, service life evaluation through a probabilistic approach is very limited since the representative evaluation method is not suggested in specifications and design code; therefore, neither the effect of the design parameter on service life nor target durability failure is not clear. The variation of design parameters also affects service life, and the allowable variation of design parameters and safety factors can be reasonably derived through a comparison of the results from the deterministic and probabilistic approach. In this study, the changing service life was analyzed through deterministic and probabilistic methods for concrete repaired with normal repair material and bacteria coating for RC sewage structures. To this end, four major design parameters such as the exterior sulfate ion concentration, diffusion coefficient, cover (coating) depth, and surface roughness, were defined. Furthermore, the effect of each parameter on the service life was evaluated and the results from the two methods were compared considering the design parameter variation. The developed bacteria coating was evaluated to be effective to extend service life for its engineering advantages and the service life variations with changing characteristics of design parameters were quantitatively analyzed in the study.

## 2. Service Life Evaluation of RC Sewage Structure with Bacteria-Coating Material with Deterministic and Probabilistic Methods

### 2.1. Governing Equation of Service Life under Sulfate Exposure

Many studies have been performed for relating a damage from sulfate ingress with service life evaluation considering the changes in the chemical composition, pH, weight and strength loss, and resistance to freezing and thawing actions [1,9,26]; however, a quantitative equation with design parameters is necessary for service life evaluation in a deterministic and probabilistic manner. Among the techniques with design parameters such as the deterioration depth method, the multi-layer diffusion method, and the strength reduction method, the deterioration velocity model, linear deterioration depth with sulfate ion diffusion, proposed by American Concrete Institute (ACI) is most widely used [1,21].

This model assumes the deterioration velocity (R) as a linear function of time considering the penetration of sulfate ions into concrete, the reaction between sulfate and aluminum hydrates, and surface expansion. This is expressed as Equation (1) below.
(1)$$R=\frac{E\cdot B^{2}\cdot c_{0}\cdot C_{E}\cdot D_{i}}{\alpha \cdot \gamma_{f}(1-\nu )}$$
where E is the modulus of elasticity of concrete (MPa), B is the linear coefficient of deformation by 1 mol of sulfate ions reacting in a unit volume (=1.8 × 10^−6^ m^3^/mol), $c_{0}$ is the concentration of sulfate ion (mol/m^3^), $D_{i}$ is the sulfate ion diffusion coefficient (m^2^/s), α is the surface roughness, $\gamma_{f}$ is the concrete failure energy (=10 J/m^2^), ν is the concrete Poisson’s ratio, and $C_{E}$ is the sulfate ion concentration reacting with ettringite (mol/m^3^).

For applying this equation to the probabilistic method, design parameters with probabilistic variation are required. With respect to Equation (1), a probabilistic service life evaluation equation as expressed in Equation (2) can be set with probabilistic design parameters. In the equation, the probability of exceeding the limit state condition during the target service life is defined and the service life is calculated based on this relationship.
(2)$$p_{f}\left[\frac{E\cdot B^{2}\cdot c_{0}(\mu ,\sigma )\cdot C_{E}\cdot D_{i}(\mu ,\sigma )}{\alpha (\mu ,\sigma )\cdot \gamma_{f}(1-\nu )}(t)>C_{d}(\mu ,\sigma )\right]>p_{d}$$
where $p_{f}(t)$ is the durability failure probability for the deterioration depth that increases with time, $C_{d}(\mu ,\sigma )$ is the probability distribution for the cover depth, and $p_{d}$ is the target durability probability assumed during durability design. In Equation (2), the exterior sulfate ion concentration ($c_{0}$), diffusion coefficient ($D_{i}$), surface roughness (α), and cover depth ($c_{d}$) were defined as random variables for design parameters.

In the case of chloride attack and carbonation, the service life limit state conditions and target durability failure probability are determined, respectively [21,30,33], but in the case of sulfate attack, there are currently no clearly set limit conditions. Therefore, as in the case of carbonation, in which the increasing carbonation depth is compared with design cover depth, the time that the deterioration depth reaches the cover depth is assumed as the limit state condition. Figure 3 shows a general overview of the probabilistic service life evaluation method, and Table 1 outlines the engineering uncertainties that require probabilistic analysis. In Figure 3, the concept of evaluating the respective deteriorating factors and design resistance is referred to as the service period concept (two lines with S(t) and R(t)), and the concept considered as one probability distribution is referred to as the lifetime concept (single line with f[R,T]), and each concept is presented in Equations (3) and (4), respectively [28].



(3)
$$P_{L,T}=P{\left[R(t)-S(t)<0\right]}_{T}<P_{\mathrm{target}}$$



(4)L=f[R,S]
here, R(t) and S(t) represent the resistance function and deterioration function with time, and $P_{\mathrm{target}}$ denotes the allowable value of the durability probability maintained during the target service life [25]. In Equation (2), since there is no prior study clearly indicating this value, $P_{\mathrm{target}}$ was assumed to be 10%, which is the same level as the case of chloride attack, and the sum of the period when the durability failure probability of the repair material reaches 10% and the durability failure probability reaches 10% at the cover depth was defined as the service life. In addition, during the probabilistic design, the MCS (Monte Carlo Simulation) was performed, and the schematic diagram of the analysis process is presented in Figure 4.

For the deterministic design method, the period in which the deterioration depth exceeds the sum of the repair material and the cover depth was considered, and the environmental factor ($\gamma_{p}$) 1.1 and durability reduction factor ($\phi_{k}$) 0.92 proposed in the specifications were adopted as shown in Equation (5) [20,33].
(5)$$\gamma_{p}Z_{p}\leq \phi_{k}Z_{\lim}$$
where $Z_{p}$ is a predicted value of the chemical erosion depth and is derived from the deterioration velocity formula of Equation (1). $Z_{\lim}$ is the design cover depth considering the construction variation.

### 2.2. Bio-Coating Materials Mixed with Bacteria

#### 2.2.1. Preparation of Repair Coating Material with Bacteria

A bacteria-coating material was developed using the culture medium in which the *Rhodobacter capsulatus* strain selected in the previous study [15] was cultured at a concentration of 10^9^ cells/mL. The mixing proportion details of the bacteria-coating material are presented in Table 2, and the detailed mixing proportions of the normal repair mortar in which the bacteria-immobilizing material is not mixed are also shown. Table 3 shows the mixing proportions of the concrete used in the existing RC sewage structure. The chemical properties of ordinary Portland cement (OPC), fly ash (FA), and ground granulated blast furnace slag (GGBFS) are shown in Table 4, and the physical properties of each material are summarized in Table 5. For the aggregates used for the preparation of bacteria-coating material, silica sand with grain diameters of 0.05–0.17, 0.17–0.25, and 0.25–0.70 mm was mixed in the same ratio for use. Figure 5 shows an overview of the improvement of the resistance performance to sulfate deterioration in the RC sewage structure with the application of the developed bacteria-coating material.

#### 2.2.2. Diffusion Coefficient of Sulfate Ion through Natural Diffusion Cell Test

In the deterioration depth evaluation, the diffusion coefficient is the governing factor, and it changes with the mixing proportions. In the previous study, the sulfate ion diffusion coefficient was evaluated using the ratio between the sulfate and chloride ion diffusion coefficients [24], but in this study, 10 diffusion cells were prepared for defining the coefficient of variation (COV). By performing the natural diffusion cell test [34], the sulfate ion diffusion coefficient of bacteria mixed coating, concrete, and normal repair material were evaluated. The equation for derivation is presented in Equation (6), and the related experimental photos are shown in Figure 6.
(6)$$D_{i}=\frac{V\Delta Q}{A\Delta T}\times \frac{L}{c_{1}-c_{2}}$$
where V is the volume of the diffusion cell (m^3^),ΔQ is the increase in sulfate ion concentration in the cell containing distilled water (kg/m^3^), A is the area of the exposure of the slice specimen (m^2^), ΔT is the elapsed time after the test starts (sec), L is the slice specimen thickness (m), $c_{1}$ is the concentration of sulfate ion solution (kg/m^3^), and $c_{2}$ is the average concentration of the cell containing distilled water (kg/m^3^).

## 3. Changing Service Life with Probabilistic Characteristics of Design Parameters

### 3.1. Service Life Evaluation with the Deterministic Method

As described above, the deterioration depth relation shown in Equation (1) is linearly proportional to the sulfate ion diffusion coefficient, and the period when the deterioration depth reaches the cover depth is calculated with an increasing period. The variables for deterministic service life evaluation are shown in Table 6. The service life of the structure was determined as the period when the deterioration depth reached the cover depth in the case without coating, and as the period when the deterioration depth reached the sum of the coating thickness and cover depth in the case with a coated structure. In addition, the cover depth of the concrete base material was set to 30 mm, and the coating thickness of the normal repair mortar and bacteria-coating material were set to 5.0 mm, which were conventionally applied. As a result of the analysis, the deterioration depth according to the exterior environment conditions is shown in Table 7, and the service life evaluation results are shown in Figure 7 and Table 8. The bacteria coating showed a much lower diffusion coefficient of 8.3–8.4% compared to normal repair mortar and sewage concrete; therefore, a considerably long service life was evaluated, even with a small coating thickness. The exterior sulfate concentration varies with the distance from the commercial facilities’ sewage outlet and 120–200 ppm of concentration level was reported [35]. In this analysis, two levels of concentration were considered, one for normal condition (120 ppm) and the other is severe condition (200 ppm), respectively.

### 3.2. Service Life Evaluation Using Probabilistic Method

#### 3.2.1. Preparation of Bacteria-Coating Material

Among the design parameters, four random variables were considered, which include cover depth, sulfate ion diffusion coefficient, surface roughness, and exterior sulfate concentration. The surface roughness and exterior sulfate concentration were assumed to be 0.1, which is a reasonable level in engineering aspect. As previously explained, the sulfate ion concentration level in RC sewage structure was 120–200 ppm [35]; however, the more severe condition of 280 ppm was added considering rapidly increasing, domestic and industrial sewage pollution. As for the thickness of the repair material and sewage RC concrete, and the diffusion coefficient, their COVs were obtained from testing 10 times, and all had variations between 0.10 and 0.14. Figure 8 shows the probability characteristics of each design parameter.

#### 3.2.2. Service Life Change of Bacteria-Coating Material


(1)Overview of service life evaluation of repair materials


In order to evaluate the durability failure probability and service life of the developed bacteria coating, the effects of the design parameters on the service life were analyzed. For the design parameters of the bacteria coating, a normal distribution and a 0.1 COV were used. An MSC was performed with the design parameter values in Table 6 as the mean values. The changes in service life were analyzed considering the changes in the mean and variation.
(2)Changes in failure probability and service life according to varying mean value

As a result of analyzing the changes in failure probability while increasing the external sulfate ion concentration at three levels from 120, 200, to 280 ppm, the service life showed a rapid decrease to 19.5, 11.2, and 8.2 years, respectively. In terms of the coating thickness, it was analyzed as the thickness increased from 5.0 mm, which is the typical coating thickness, to 15 mm. The service life was evaluated to be 11.2, 22.2, and 33.9 years, respectively, indicating more a significant impact compared to the other parameters. The surface roughness showed a wide range of values (1–10) in previous studies [4,36], and the failure probability was analyzed with a changing surface roughness in the range of 1.5–3.5. The failure probability decreased with an increasing surface roughness, and the service life accordingly increased to 11.2, 18.8, and 26.1 years, respectively. The diffusion coefficient is linearly proportional to the deterioration depth and, since it varies widely depending on the mixing proportion, it is an important design parameter. The failure probability and service life were derived by changing the sulfate ion diffusion coefficients derived through the natural diffusion cell test to 0.5, 1.0, and 2.0 times, and the service life showed a rapid decrease to 22.1, 11.2, and 5.4 years, respectively.

Figure 9 shows the changes in durability failure probability for each service life evaluation, and the service life corresponding to the target durability probability of 10% is plotted in Figure 10, which shows service life from the probabilistic method.

Based on the design constants for actual construction, if the change ratio with reference to the baseline value (average) of each design parameter can be illustrated, the effect of each parameter on the service life can be quantitatively evaluated. The changes in service life gradient with respect to the baseline conditions (exterior sulfate concentration, 200 ppm; coating thickness, 5.0 mm; surface roughness, 1.5; and $D_{i}$, 1.0) are depicted in Figure 11. The exterior sulfate concentration and diffusion coefficient show almost the same effect, and surface roughness and coating thickness have a relatively large effect. That is, surface roughness has a close relationship with the strength between the aggregate and the base concrete, and cover depth is also a primary mechanism to block the direct deteriorating factors from the outside, and therefore, the careful determination of the two design parameters is required for securing performance during service life.
(3)Failure probability and changes in service life according to COV

In this section, the variation in the service life is evaluated while increasing the COV of each design parameter from 0.1 to 0.3. Even with the change in COV, the period with a failure probability at 50% is evaluated to be the same result from the deterministic method under the assumption of normal distribution. In other words, in the domain before the failure probability at 50%, the larger COV causes the shorter service life, and therefore, reducing their COV, considering the construction and material quality, is instrumental. Figure 12 shows the changes in the failure probability of each design parameter with changing COV. When the COV of one variable increased, the COV of the other design parameters was all fixed to 0.1 for evaluation. The changes in service life derived according to the variation of design parameters are shown in Figure 13. When the COV increases from 0.1 to 0.3, the service life of the coating decreased by 2.5 to 2.6 years, amounting to a 13–22% reduction in service life. Therefore, reducing the variation in quality during construction is highly important.

## 4. Service Life Changes in RC Sewage Structure with Different Repair Materials

In this section, the service life was evaluated with two exterior sulfate concentrations, a normal sulfate ion level (120 ppm) and a slightly higher level (200 mm) referred to the previous field assessment [35]. As previously explained, the service life was defined to be the period when the deterioration depth exceeds the sum of the cover depth and coating thickness, and it was evaluated with the following three cases: concrete only without repair material, protection with normal repair mortar, and protection with a bacteria coating. In addition, the service life of the structure was evaluated by assigning each design parameter a variation of 0.1, 0.2, and 0.3.

Figure 14 shows the changes in the service life of the RC sewage structure with the exterior sulfate concentration. In the deterministic method, the service life was evaluated as 21.6 years in the case without repair under the environmental exposure to 120 ppm, it was 25.7 years when a normal repair mortar was applied, and 42.1 years when a bacteria-coating material was applied. When the probabilistic method was adopted, the service life was slightly reduced. When the COV increased from 0.1 to 0.3, the service life showed a sharp decrease from 26.5 to 15.0 years for normal concrete, from 30.6 to 17.2 years when the normal repair mortar was used, and from 45.1 to 25.3 years when bacteria-coating was applied (Figure 15). The reduction rate was evaluated to be between 56.1 and 56.6%, and in the case of 200 ppm, the reduction rate was from 56.2 to 57.7%, indicating a similar level. In the case of a sulfate concentration of 200 ppm, with the deterministic method, the service life was evaluated at 12.9 years without repair; this increased to 15.4 years when normal repair mortar was applied, and further increased to 25.3 years when a bacteria coating was used. When the probabilistic method was applied in the same environment, the service life was in the range of 9.2–16.0, 10.5–18.2, and 15.4–27.4 years, respectively, depending on the variation, showing a significant difference depending on the type of repair material.

The smaller the coating thickness of the normal RC sewage structure is, the less clogging there will be in the flow. Considering the coating thickness of 5 mm, the bacteria coating demonstrated a 1.47–1.50 times longer service life than normal repair mortar, and the service life was evaluated to be longer as the COV decreased. In the case of the developed bacteria mixed coating, it was shown that the quality can be maintained steadily under the influence of continuous sulfide ions and humidity. The service life of an RC sewage structure repaired with a bacteria-coating material under sulfate ion ingress can be extended considerably due to a low diffusion coefficient and low COV.

Considering the conventional unit repair cost and intended service life (60 years), the repair cost can be reduced due to extended service life and decreasing the number of repair events. The unit repair cost [37] and total repair cost under 120 and 200 ppm of sulfate ingress are shown in Table 8 and Figure 16. As shown, the repair events were reduced from nine to one time and the repair cost was reduced from 441.9 to 65.8 USD/m^2^ under 120 ppm. For the 200 ppm case, they were reduced from 19 to 3 times and 932.9 to 98.7 USD/m^2^, respectively.

**Table 8 materials-14-05424-t008:** Unit repair cost for each technique (thickness = 5 mm).

	Repair Mortar	Bacteria-Coating Material
Repair cost (USD/m^2^)	49.1	32.9

## 5. Conclusions

In this study, a repair coating technique with bacteria was developed and the service life of an RC sewage structure under sulfate ion ingress was evaluated with the following two different methods: the deterministic and probabilistic method. The conclusions from this study are summarized as follows:An FA-GGBFS-based repair material with *Rhodobacter capsulatus* was developed, which consumes sulfate ions as nutrients. For applying probabilistic service life evaluation, probability distribution characteristics were derived through diffusion cell tests and cover depth measurements. The derived diffusion coefficient of the developed repair material with *Rhodobacter capsulatus* showed a very low value in the level of 8.3–8.4% compared to that of the base concrete and normal repair material, and the COV was around 10%, indicating stable quality assurance.Five major design parameters (surface coating, coating thickness, diffusion coefficient, exterior sulfate ion concentration, and surface roughness) were defined as random variables for analyzing the service life with a probabilistic approach. The effect of the mean value of surface roughness and cover depth was evaluated to be dominant; therefore, it is important to pay special attention to the surface treatment and coating thickness during repair construction. When the service life was evaluated while increasing the COV of each design parameter from 0.1 to 0.3, the service life was reduced by 1.5 to 2.6 years, even with the application of the appropriate coating thickness, which yields a decrease in service life by more than 15%.When the service life of the RC sewage structure was evaluated considering under 120 ppm of exterior sulfate ingress, it was evaluated to be 21.6 years without repair, but this increased to 25.7 years with a normal repair mortar, and further increased to 42.1 years enhanced with a bacteria-coating material. When the probabilistic method was applied, it was reduced due to the low target durability probability. With the increasing COV of the design parameter, it was greatly reduced from 26.5 to 15.0 years for base concrete, from 30.6 to 17.2 years for normal repair mortar, and from 45.1 to 25.3 years with the developed bacteria-coating application. The service life reduction rates were evaluated to be around 56.1–56.6% (120 ppm of sulfate ion), and 56.2–57.7% (200 ppm of sulfate ion). Even when the applied coating thickness of the sewage structure is thin (5 mm), the bacteria-coating shows a 1.47–1.50 times longer service life than that with the normal repair mortar.Utilizing the unit repair cost and applying the developed bacteria repair technique reduced the total repair cost from 441.9 to 65.8 USD/m^2^ (120 ppm) and from 932.9 to 98.7 USD/m^2^ (200 ppm), respectively. It is because of the low diffusion coefficient of the bacteria repair material and low unit repair price.The developed repair material with *Rhodobacter capsulatus* consumes sulfide ions as nutrients; therefore, it can assure a continuous maintenance of quality level; however, further research is still required on the rapid increase in flow velocity due to flooding, damage to collisions with floats, temperature management and quality control during coating composite manufacturing, and temperature effects during construction.

## Figures and Tables

**Figure 1 materials-14-05424-f001:**
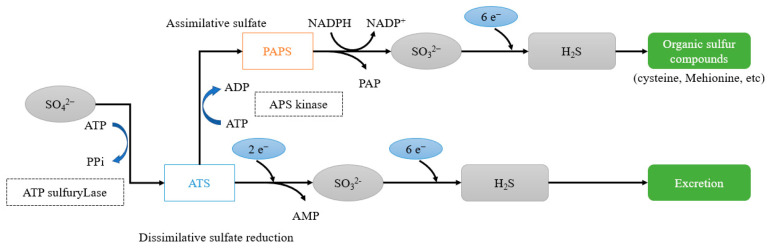
Metabolic reaction process of sulfate reducing bacteria.

**Figure 2 materials-14-05424-f002:**
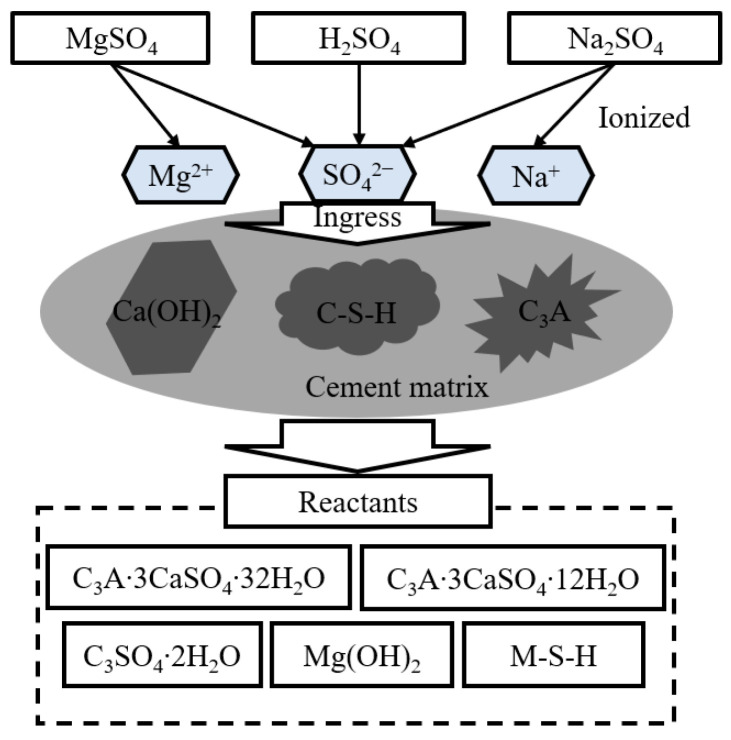
Deterioration mechanism of cement matrix exposed to sulfate ion ingress.

**Figure 3 materials-14-05424-f003:**
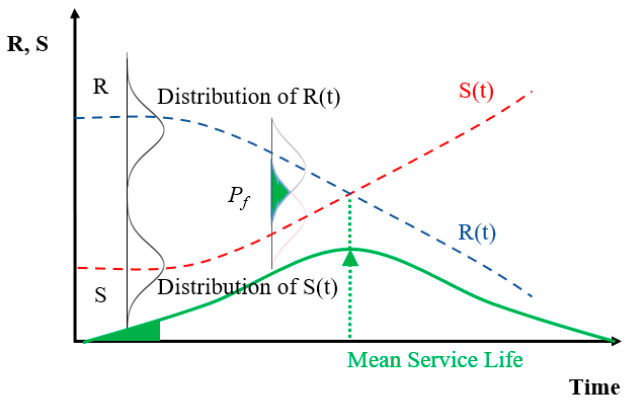
Durability design of probabilistic model with service life and lifetime concept.

**Figure 4 materials-14-05424-f004:**
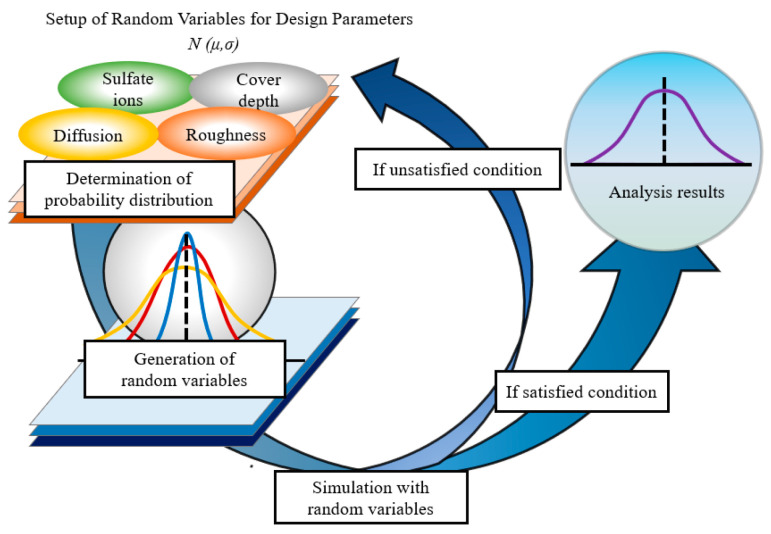
Analysis step of MCS with random variables of design parameters.

**Figure 5 materials-14-05424-f005:**
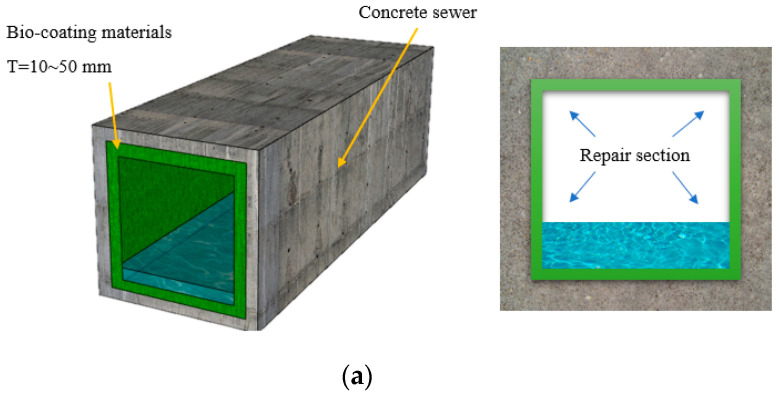
Schematic design for repair with bacteria and RC sewage structure: (**a**) RC culvert with bacteria coating repair; (**b**) Metabolism of resistance to sulfate attack by bacteria-formed slime.

**Figure 6 materials-14-05424-f006:**
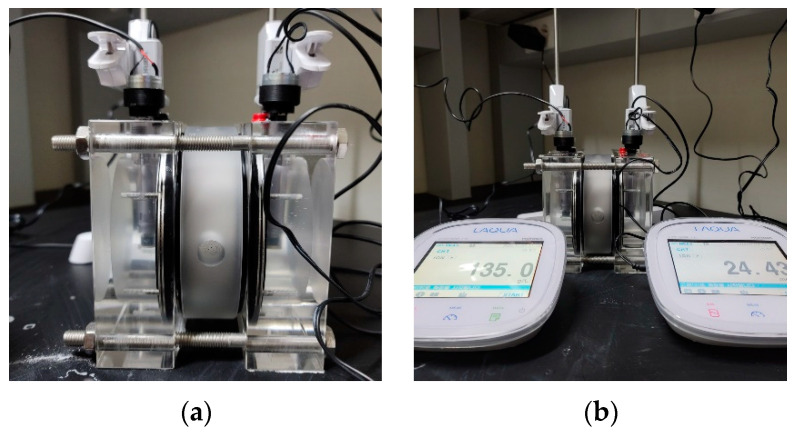
Photo for sulfate ion diffusion test: (**a**) setting image of test specimens; (**b**) setting image of ion meter for recoding the chaining sulfate ion concentration.

**Figure 7 materials-14-05424-f007:**
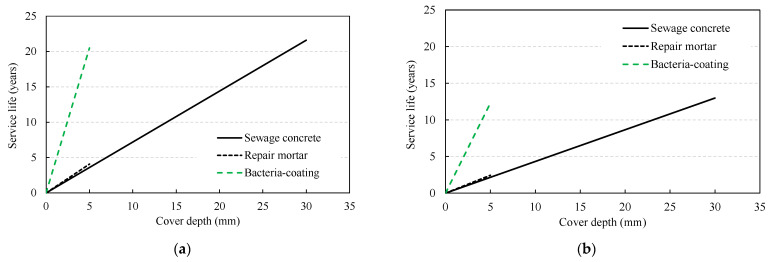
Service life evaluation through deterministic approach: (**a**) sulfate concentration = 120 ppm (normal condition); (**b**) sulfate concentration = 200 ppm (severe condition).

**Figure 8 materials-14-05424-f008:**
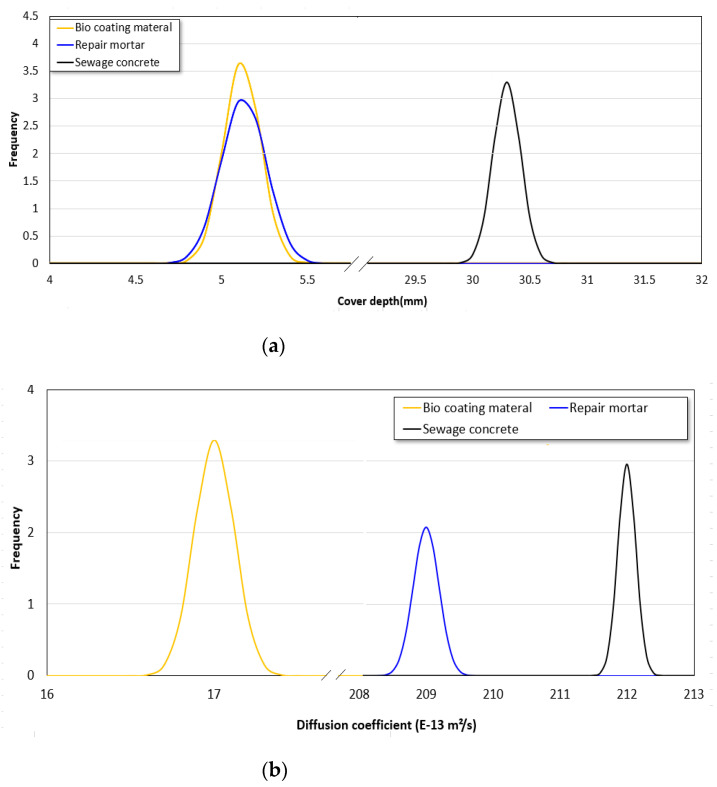
Probability distributions of design: (**a**) probabilistic distribution of thickness; (**b**) probabilistic distributions of diffusion coefficient.

**Figure 9 materials-14-05424-f009:**
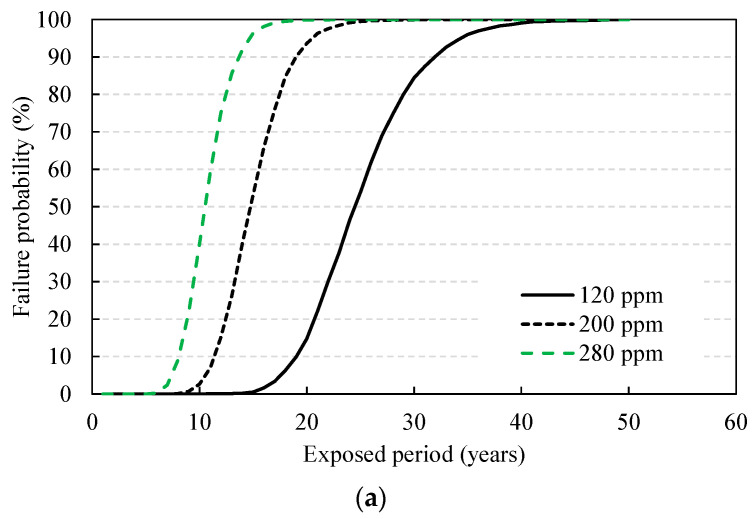
Changes in failure probability with design variables: (**a**) exterior sulfate ions; (**b**) coating thickness; (**c**) surface roughness; (**d**) diffusion coefficient.

**Figure 10 materials-14-05424-f010:**
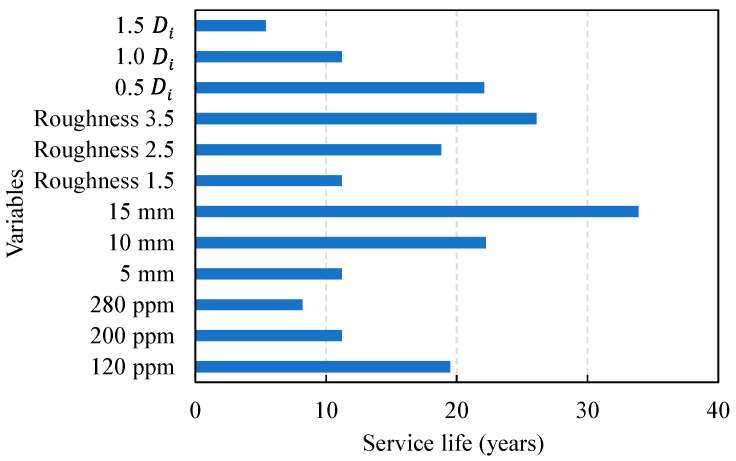
Changes in service life with design variables.

**Figure 11 materials-14-05424-f011:**
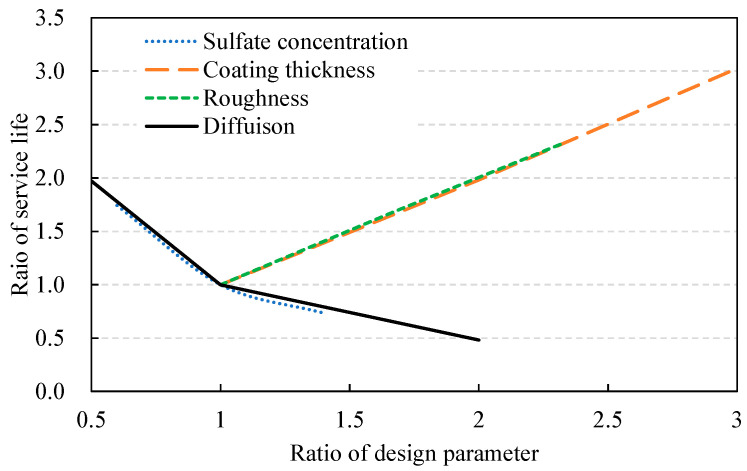
Relationship between design parameters and service life.

**Figure 12 materials-14-05424-f012:**
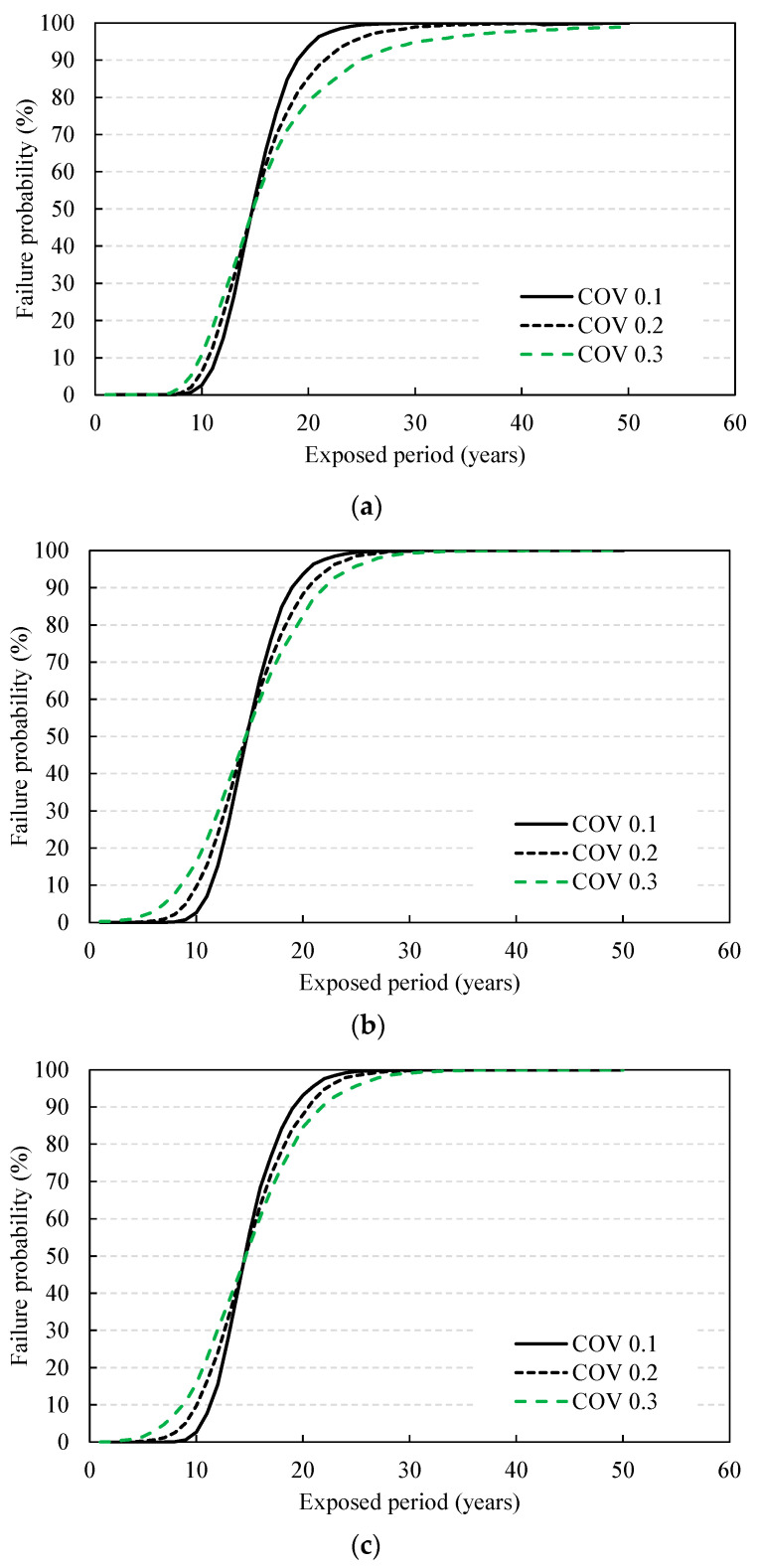
Changes in failure probability with COV variation: (**a**) exterior sulfate ions; (**b**) coating thickness; (**c**) surface roughness; (**d**) diffusion coefficient.

**Figure 13 materials-14-05424-f013:**
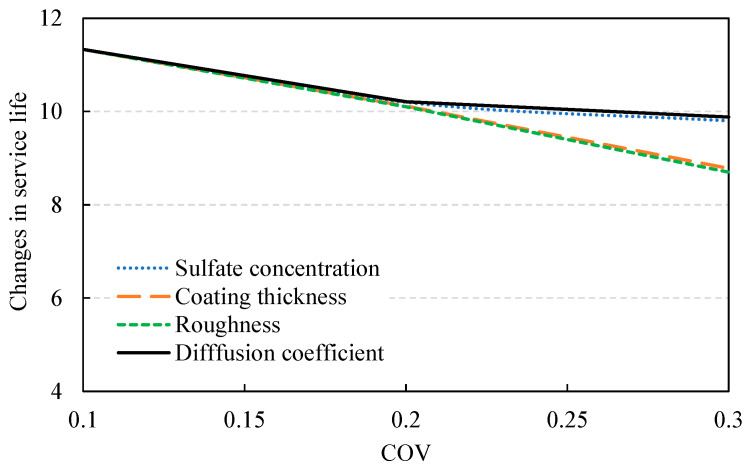
Decreasing service life with increasing COV.

**Figure 14 materials-14-05424-f014:**
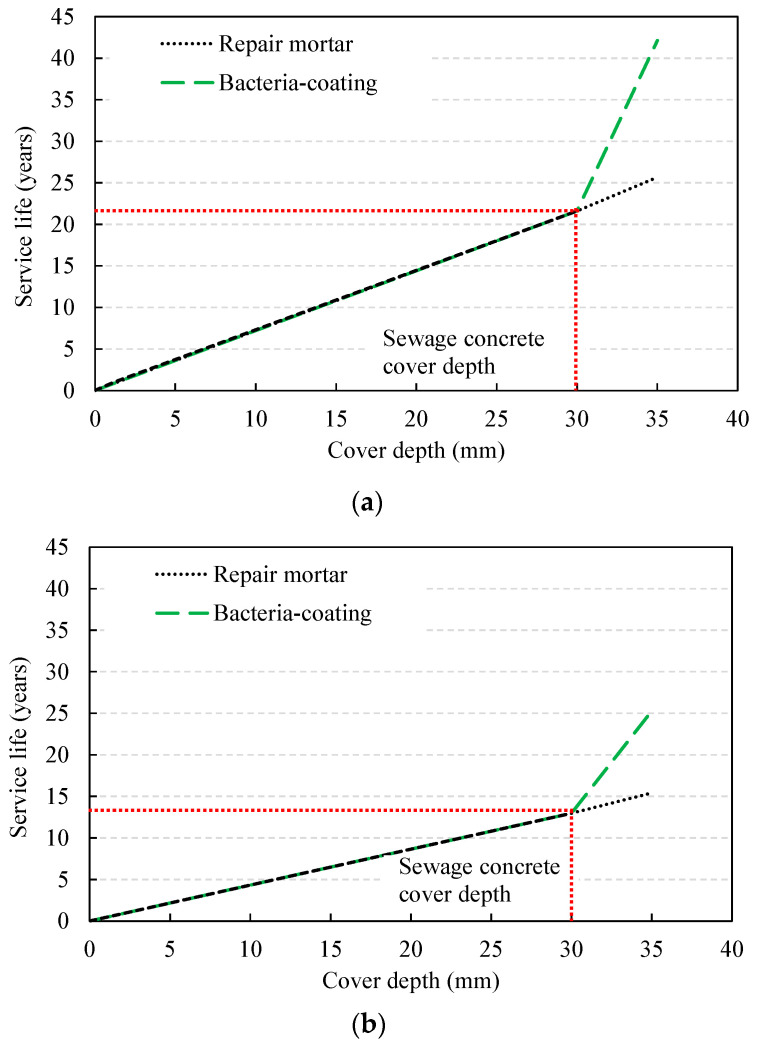
Service life variation according to deterministic method: (**a**) sulfate concentration = 120 ppm; (**b**) sulfate concentration = 200 ppm.

**Figure 15 materials-14-05424-f015:**
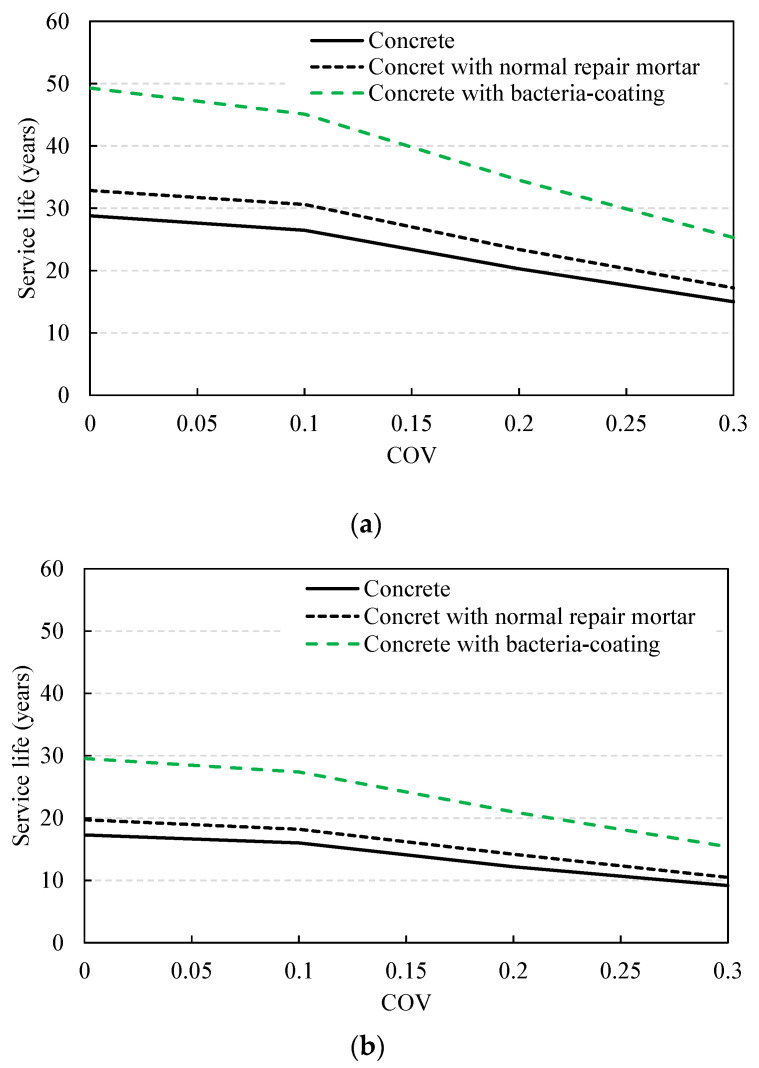
Service life variation with increasing COV according to probabilistic method: (**a**) sulfate concentration = 120 ppm; (**b**) sulfate concentration = 200 ppm.

**Figure 16 materials-14-05424-f016:**
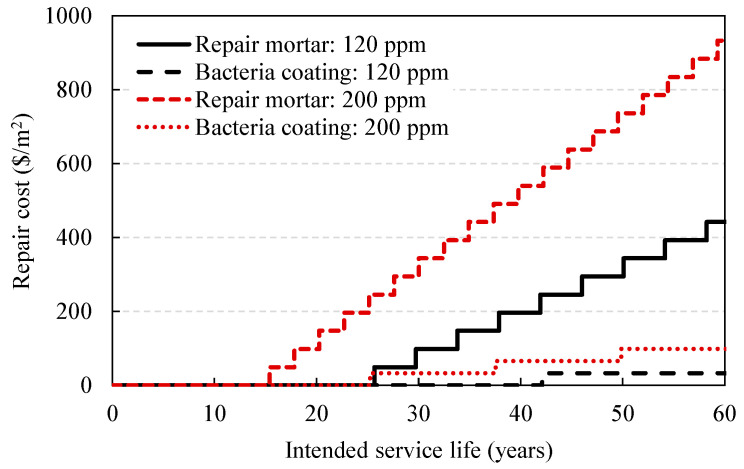
Variation in repair cost due to increase in service life with normal repair mortar and bacteria-coating material.

**Table 1 materials-14-05424-t001:** Engineering uncertainties in durability design.

Uncertainty Type	Limitation
Physical	Inherent random nature of a basic variables - Concrete cover depth - Concentration of exterior sulfate ions - Quality of concrete (diffusion coefficient of sulfate ion) - Local condition (cracks and joints)
Statistical	Assumption for probability density functions - Limited sample size
Model	Governing mechanism for deterioration by sulfate - Simplified equation of deterioration by sulfate without considering sulfuric reaction - Assumption of material properties (sulfurictable and reactant material) - Assumption of non-correlated variables
Decision	Definition of durability failure criteria - The period that carbonation depth exceeds the cover depth

**Table 2 materials-14-05424-t002:** Mixing proportions of repair mortar and bacteria-coating material.

Specimens	*W/B* (%)	*S/B*	Binder Mixing Ratio (%)			Replacement Ratio of Bacteria Carrier by the Volumetric Fraction of Sand (%)
OPC	FA	GGBFS
Repair mortar	35	2	35	20	45	-
Bacteria-coating material	35

**Table 3 materials-14-05424-t003:** Mixing proportion of sewage concrete.

*W/C* (%)	*S/a* (%)	Unit Weight (kg/m^3^)			
Water	Cement	Sand	Gravel
45	47	180	400	786	903

**Table 4 materials-14-05424-t004:** Chemical properties of OPC, FA, and GGBFS.

Materials	Composition (wt.%)									
SiO_2_	Al_2_O_3_	Fe_2_O_3_	CaO	MgO	SO_3_	TiO_2_	Na_2_O	K_2_O	LOI
OPC	21.7	5.3	3.1	62.4	1.6	1.7	-	-	-	0.8
GGBFS	33.5	15.2	0.5	45.6	2.6	2.5	0.9	0.2	0.4	0.7
FA	53.3	27.9	7.8	6.8	1.1	0.8	-	0.6	0.8	0.9

**Table 5 materials-14-05424-t005:** Physical properties of the used cementitious materials.

Materials	Density (g/cm^3^)	Fineness (cm^2^/g)
OPC	3.15	3260
GGBFS	2.94	4355
FA	2.23	3720

**Table 6 materials-14-05424-t006:** Analysis conditions on service life evaluation (Deterministic method).

Variables	Sewage Concrete	Repair Mortar	Bacteria-Coating Material
$c_{0}$	200	200	200
$D_{i}$	2.12 × 10^−12^	2.09 × 10^−12^	0.17 × 10^−12^
E	25,700	21,500	21,500
ν	0.17	0.27	0.27
α	1.5	1.5	1.5
$\gamma_{f}$	10	10	10
Binder weight	400	300	300
$C_{E}$	207	196	462
Cover depth (mm)	30	5	5

**Table 7 materials-14-05424-t007:** Service life through deterministic technique.

Exterior Condition (Sulfate Concentration)	Deterioration Velocity (mm/Year)		
Sewage Concrete	Repair Mortar	Bacteria-Coating Material
120 ppm	1.16	1.03	0.20
200 ppm	1.94	1.71	0.34

## Data Availability

The data presented in this study are available on request from the corresponding author.

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
