# Peer review of "Service Life Evaluation for RC Sewer Structure Repaired with Bacteria Mixed Coating: Through Probabilistic and Deterministic Method"

_materials, 2021, doi:10.3390/ma14185424_

Round 1

Reviewer 1 Report

Dear Authors,

This paper is describing cracking and evaluation on RC sewer structure. Results are very interesting and quality of the paper presentation is quite good.  I propose the paper goes through a minor revision and the below comments needs to be addressed in order to improve the quality of the paper.

Comments:

1- State of the Art needs to be improved and include different cracks and then include different NDT which is used for evaluation and detection of cracks. For example, GPR and some more?

NDT assessment of rigid pavement damages with Ground Penetrating Radar: Laboratory and field tests. International Journal of Pavement Engineering, Taylor and Francis, 2020. https://doi.org/10.1080/10298436.2020.1778692

2- All figures need re-design and please consider the description and indications of each part of the sketch.

3- I suggest add a section explain the advantages and limitations of your proposed approach.

4- Please, re-write the conclusion and reflect on your results.

5- Figures 5 and 6 need to be clear and differentiate with a and b orders. Use grid system background for graphs.

Looking forward to receiving the revised version.  

Reviewer 2 Report

Surface protective coating mixed with slime forming bacteria was prepared, and the changing service life and influencing design parameters of RC sewage 375 structure with the developed coating material were analyzed with deterministic and probabilistic methods. This work is good and it is one of the main areas of research for water utilities in determining the pipe's service life and reducing the maintenance cost. There are some major issue that the authors need to address in this work: 

  1. Mention what is RC on the first occurrence in the paper. General readers may get confused between reinforced concrete or Rhodobacter capsulatus.
  2. The corrosion or concrete degradation in sewers is due to microbial activities, which has very clearly mentioned in the introduction section of the manuscript. However, the microbial production of acid is highly dependent on the sewer concrete surface temperature and moisture condition. Also depending on the surface environmental conditions, there are proven data analytic models to estimate the useful service life. Refer and include the following relevant papers in the introduction for a comprehensive literature review. (a) K. Thiyagarajan, S. Kodagoda, R. Ranasinghe, D. Vitanage and G. Iori, "Robust Sensor Suite Combined With Predictive Analytics Enabled Anomaly Detection Model for Smart Monitoring of Concrete Sewer Pipe Surface Moisture Conditions," in IEEE Sensors Journal, vol. 20, no. 15, pp. 8232-8243, 1 Aug.1, 2020, doi: 10.1109/JSEN.2020.2982173. (b) Thiyagarajan, K., Kodagoda, S., Ranasinghe, R., Vitanage, D. and Iori, G., 2018. Robust sensing suite for measuring temporal dynamics of surface temperature in sewers. Scientific reports8(1), pp.1-11.
  3. Page 10: Please justify the reason why the external sulfate ion concentration at 3 levels from 120 ppm, 200 ppm, and 280 ppm was chosen. Also, does the coatings wont work in low ppm? Also, the degradation is only dependent on ppm?
  4. The manuscript has missed the key analytical and data analytical model for service life estimation. Some of the key works in the literature are: 

        Wells, T. and Melchers, R.E., 2014. An observation-based model for corrosion of concrete sewers under aggressive conditions. Cement and Concrete Research, 61, pp.1-10.

    Any effects of pH on the coating service life:

        Valix, M., Zamri, D., Mineyama, H., Cheung, W.H., Shi, J. and Bustamante, H., 2012. Microbiologically induced corrosion of concrete and protective coatings in gravity sewers. Chinese Journal of Chemical Engineering, 20(3), pp.433-438.

Reviewer 3 Report

The reviewer appreciates the valuable work done by the authors. The technical contents of the paper are in general interesting. The corresponding findings are useful for the scientific society. The paper fits properly the scopes of the “Materials, MDPI”. Nevertheless, the current state of knowledge relating to the manuscript topic has not been covered and clearly presented, and the authors’ contribution and novelty are not emphasized. In this regard, the authors should make their effort to address these issues, by adding additional comments on the state of the art and the proposed aspects. Moreover, objectives and findings should be presented more clearly. Additional comments should also be added in regard to the practical value of this research, how the industry could profit from this research. I do not recommend the publication of the manuscript in the “Materials, MDPI” unless the aforementioned corrections are done.

Round 2

Reviewer 2 Report

The authors have addressed the reviewers concerns appropriately. Recommend this article for publication.

Reviewer 3 Report

The authors carried out the required revisions.